# *Ancylostoma ceylanicum*: The Neglected Zoonotic Parasite of Community Dogs in Thailand and Its Genetic Diversity among Asian Countries

**DOI:** 10.3390/ani10112154

**Published:** 2020-11-19

**Authors:** Doolyawat Kladkempetch, Sahatchai Tangtrongsup, Saruda Tiwananthagorn

**Affiliations:** 1Master’s Degree Program in Veterinary Science, Faculty of Veterinary Medicine, Chiang Mai University, Chiang Mai 50100, Thailand; doolyawat_kl@cmu.ac.th; 2Department of Companion Animal and Wildlife Clinic, Faculty of Veterinary Medicine, Chiang Mai University, Chiang Mai 50100, Thailand; sahatchai.t@cmu.ac.th; 3Research Center of Producing and Development of Products and Innovations for Animal Health and Production, Faculty of Veterinary Medicine, Chiang Mai University, Chiang Mai 50100, Thailand; 4Department of Veterinary Biosciences and Veterinary Public Health, Faculty of Veterinary Medicine, Chiang Mai University, Chiang Mai 50100, Thailand; 5Excellent Center in Veterinary Bioscience, Faculty of Veterinary Medicine, Chiang Mai University, Chiang Mai 50100, Thailand

**Keywords:** *Ancylostoma ceylanicum*, community dogs, ITS region, *cox1*, Thailand, population diversity

## Abstract

**Simple Summary:**

Ancylostomiasis is a zoonotic disease caused by the *Ancylostoma* hookworm infection of dogs, cats, and other wildlife species and is frequently found in Asia and tropical regions. The present study had confirmed the species of hookworm in dogs and soil environments collected from temple communities. In addition, we investigated the association of hookworm-contaminated soil in temple areas to increase awareness and establish a regimen to prevent zoonotic hookworm transmission. Lastly, we analyzed the genetic diversity and evolution of hookworm in dogs and soil environments among Thai *A. ceylanicum* and other Asian populations and disclosed insight into their fundamental genetic relationship.

**Abstract:**

*Ancylostoma ceylanicum* is a zoonotic helminth that is commonly found in domestic dogs and cats throughout Asia but is largely neglected in many countries. This study aimed to confirm the species of hookworm in dogs and soil environments and investigate the evolutionary analyses of *A. ceylanicum* among Thai and Asian populations. In a total of 299 dog fecal samples and 212 soil samples from 53 temples, the prevalence rates of hookworm infection by microscopic examination were 26.4% (79/299) and 10.4% (22/212) in dog and soil samples, respectively. A PCR-RFLP targeting the ITS region was then utilized to identify the hookworm species. In dogs, *A. ceylanicum* was the main hookworm species, and the rates of *A. ceylanicum* and *A. caninum* infections were 96.6% and 3.5%, respectively. The genetic characterization and diversity indices of the *A. ceylanicum*
*cox1* gene among Thai and Asian populations were evaluated. Nine haplotypes were identified from Thai *A. ceylanicum*, in which the haplotype diversity and the nucleotide diversity were 0.4436 and 0.0036, respectively. The highest nucleotide diversity of Chinese *A. ceylanicum* populations suggested that it could be the ancestor of the populations. Pairwise fixation indices indicated that Thai *A. ceylanicum* was closely related to the Malaysian population, suggesting a gene flow between these populations. The temples with hookworm-positive dogs were associated with the presence of hookworm-contaminated soil, as these levels showed an approximately four-fold increase compared with those in temples with hookworm-negative dogs (OR = 4.38, 95% CI: 1.55–12.37). Interestingly, the genotypes of *A. ceylanicum* in the contaminating soil and infecting dogs were identical. Therefore, increased awareness and concern from the wider public communities with regard to the responsibility of temples and municipal offices to provide proper deworming programs to community dogs should be strongly endorsed to reduce the risk of the transmission of this zoonotic disease. In addition, parasitic examination and treatment should be strongly implemented before dogs are imported and exported worldwide.

## 1. Introduction

Hookworms are soil-transmitted gastrointestinal nematodes of the Ancylostomatidae family that infect dogs, cats, and humans. These nematodes are endemic in tropical and subtropical regions of the world, including parts of China, Southeast Asia, Australia, and Africa [1,2,3,4]. Hookworms can infect the host by the fecal-oral route and through the skin penetration of the third-stage larvae [2]. Hookworm infection causes intestinal blood loss, anemia, malnutrition, and dermatitis [5]. Common species of hookworm infection in dogs include *Ancylostoma caninum*, *A. ceylanicum*, *A. braziliense*, and *Uncinaria stenocephala*, while *A. braziliense*, *A. tubaeforme*, *A. ceylanicum*, and *U. stenocephala* are commonly found in cats [6,7]. Recent studies have shown that ancylostomiasis in humans results from *Necator americanus*, *A. duodenale*, and *A. ceylanicum* [5,8,9]. Among *Ancylostoma* species, *A. ceylanicum* is the only species of animal hookworms known to produce patent infections in humans and is an important cause of zoonotic ancylostomiasis in Asia and Southeast Asia [9,10,11,12,13,14]. 

In general, the epidemiological study revealed that animal-derived *Ancylostoma* species are emerging due to human and animal interaction [15,16]. In Thailand, free-roaming community dogs have been widespread and come in close contact with people in some areas, such as temples and rural communities. The presence of hookworm larvae in soil reported by George et al. [5] confirmed that humans and dogs are at risk of infection from hookworm-contaminated soil. Additionally, a previous report in Malaysia from Ngui et al. [15] indicated that people who are not wearing shoes are at risk of hookworm infection from hookworm-contaminated soil. The opportunity for exposure to hookworm is high in children, farmers, and especially Buddhist monks, who do not wear shoes based on their practices and Buddhist tradition in Thai temples. Information regarding whether the soil in the area is contaminated with hookworm is, therefore, important for public health to promote awareness and establish a regimen to prevent zoonotic hookworm transmission.

The conventional method for hookworm diagnosis is coprological examination. However, this method cannot differentiate hookworm species due to the similar egg size and morphology. Morphological identification of adult hookworms is achievable, but the process is time-consuming and labor-intensive and requires personal skill [16]. Alternatively, molecular techniques have been developed for the identification of hookworms at the species level, such as polymerase chain reaction (PCR) assays and PCR with restriction fragment length polymorphism (PCR-RFLP) assays targeting the internal transcribed spacer (ITS) region [16]. The ITS region is a conserved region with a low mutation rate and has no intraspecific variation [17]. Thus, the ITS region is suitable as a genetic marker for hookworm species identification [18]. Recently, the cytochrome *c* oxidase subunit 1 (*cox1*) gene has been utilized to evaluate the evolution and relationship of hookworm species among the population [19,20], given its high intraspecific sequence variation from maternal inheritance and high evolutionary rates [21]. 

In Northern Thailand, a few reports have assessed the prevalence of hookworm infection in humans, domestic dogs, and cats. By microscopic examination, hookworm infection rates of approximately 21.3% in dogs and 13.9% in cats [22] and 12% in dogs and 4.5% in cats [23] have been reported in the lower northern region and in the Chiang Mai province of the upper northern region, respectively. The prevalence of hookworm infection in humans ranged from 0.6% to 13.4% [24,25]. However, the species of hookworm in Upper Northern Thailand have never been clarified. The present study, therefore, aimed to identify the species of hookworm in dogs and investigate the presence of hookworm contamination in the soil environment of community temples where there are high risks of communicable pathogen transmission. Furthermore, the genetic characteristics of *Ancylostoma* hookworm and the evolutionary genetic relationships among the Thai hookworm and the populations of neighboring Asian countries have been elucidated.

## 2. Materials and Methods

### 2.1. Study Area and Sample Size

The study area is located in the upper northern region of Thailand. Four provinces, including Chiang Mai (18°47′26″ N, 98°59′14″ E), Chiang Rai (20°01′05″ N, 99°40′22″ E), Lampang (18°17′21″ N, 99°29′26″ E), and Phayao (19°11′30″ N, 99°52′47″ E), were chosen and considered as high dog population density provinces (Figure 1). Sampling sites were 53 community temples that were chosen based on convenience and distribution from at least 3 districts in each province. The inclusion criteria for temple selection were (i) a human population density of more than 30 persons per sq. km. [26,27], (ii) no regular deworming program in dogs within 3 months, and (iii) ease of access and sample transportation to the laboratory. The sample size of 299 dog fecal samples for the detection of hookworm infection was calculated using the Win episcope 2.0 program [28] based on an estimated hookworm prevalence of 12.8% in dogs [29] and an error rate of 5% with a 95% level of confidence. The stratified sampling method was used to generate the number of samples per province based on dog population data [30].

All the owners signed an informed consent form, and the animal use was approved by the Ethics Committee of the Laboratory Animal Center, Faculty of Veterinary Medicine, Chiang Mai University (S41/2561), on 18 January 2019.

### 2.2. Sample Collection and Examination

#### 2.2.1. Fecal and Soil Sample Collection

Samples were collected between June and September 2019. Two hundred and ninety-nine fecal samples were collected from the rectum of dogs or fresh fecal samples found on the ground. All the collected fresh feces were normal in appearance (medium to dark brown in color, soft to firm in texture) based on the WALTHAM fecal scoring system [31]. The dog ID, date of collection, temple name, district, province, cleaning housing program, and type of food were recorded. Soil environment samples were collected at the temple in the morning (08:00–10:00). For soil sample collection, the sites of sample collection were defined by 4 areas in each temple: (i) temple courtyard, (ii) dog dwelling area, (iii) human activity area (such as sand pagoda), and (iv) under the big tree. In the collection site that was not dirt ground, the soil sample for that site was obtained from the adjacent ground within a 2 m radius of the area. Soil samples were collected from moist areas within the selected sampling sites, and approximately 50 g (wet weight) was obtained by scraping the surface layer to gather a 0.09 square meter area using a metal spade. The soil samples were kept separately in a sealed plastic bag. 

All the samples were stored on ice and transported to the Parasitology Laboratory, Faculty of Veterinary Medicine, Chiang Mai University. Samples were refrigerated at 4 °C upon arrival.

#### 2.2.2. Microscopic Examinations of Fecal and Soil Samples 

Fresh fecal samples were examined under a light microscope after simple floatation using zinc sulfate solution (specific gravity: 1.18) for the presence of eggs. The remainder of the fecal sample was stored at −20 °C until further molecular analysis. The Baermann technique was used to determine the presence of hookworm-like larvae in the soil samples [32]. Twenty grams of soil samples were primarily filtered through a tea strainer to remove any debris and then subjected to another filtration step using 2 sieve filters with pore sizes of 0.4 and 0.2 mm. The filtrated soil sample was cultured individually using tap water. Hookworm-like larvae containing sediments were stored at −20 °C until further molecular analysis could be carried out.

### 2.3. DNA Extraction

Genomic DNA (gDNA) samples from hookworm-positive samples were extracted using the NucleoSpin^®^ DNA stool kit (Macherey-Nagel GmbH, Düren, Germany) according to the manufacturer’s instructions. In the elution step, gDNA samples were eluted with 100 μL of elution buffer. The extracted DNA concentration and quality were determined using a spectrophotometer (Beckman Coulter DU^®^ 730 Life Sciences, Pasadena, CA, USA) at wavelengths of 260 and 280 nm. All the gDNA samples were kept at −20 °C until the next step.

### 2.4. Molecular Analyses

#### 2.4.1. Identification of Hookworm Species Using a Polymerase Chain Reaction-Restriction Fragment Length Polymorphism (PCR-RFLP) Assay Based on ITS Region

The PCR-RFLP assay, which was established by Traub et al. [16], was conducted to identify the species of hookworm in the positive dog fecal and soil samples. A forward primer RTGHF1 (5′-CGTGCTAGTCTTCAGGACTTTG-3′) and a reverse primer RTABCR1 (5′-CGGGAATTGCTATAAGCAAGTGC-3′) were used to amplify the 545-bp region of the internal transcribed spacer (ITS1, 5.8S, and ITS2). PCR amplification reactions were performed in a 25 μL reaction volume containing 10 ng of template gDNA (2–4 μL), 0.2 μM of each primer (0.5 μL of 10 μM), and 12.5 μL of 2× PCR Master Mix (KOD One™ PCR Mastermix; Toyobo, Japan). The thermocycling conditions were as follows: initial denaturation at 94 °C for 2 min, 45 cycles of amplification (94 °C for 30 s, 62 °C for 30 s, and 72 °C for 30 s), and a final extension at 72 °C for 5 min. The reaction was performed on a T100™ Thermal Cycler (Bio-Rad, Hercules, CA, USA). The amplified PCR products were digested with Hinf1 to differentiate *A. caninum* from *A. ceylanicum* and *U. stenocephala*. The digestions were performed at 37 °C for 16 h, using 1 unit of restriction enzyme and 1 μg of DNA (3–5 μL of PCR product) in a total reaction volume of 20 μL. The PCR products were analyzed using 2% agarose gels electrophoresis. Distilled water (DW) and the gDNA of *A. caninum* and *A. ceylanicum* were used as a negative and positive control, respectively. 

#### 2.4.2. PCR Assay and Nucleotide Analysis Based on *A. ceylanicum* and *A. caninum cox1* Gene

The mitochondrial *cox1* gene served as the target to assess the genetic characteristics and evolutionary analysis. The previously identified *A. ceylanicum-* and *A. caninum*-positive samples were successively subjected to PCR for the *cox1* gene using a set of primers described by Inpankaew et al. [33], including the forward primer AceyCOX1F (5′-GCTTTTGGTATTGTAAGACAG-3′) and the reverse primer AceyCOX1R (5′-CTAACAACATAATAAGTATCATG-3′), amplifying a 377-bp amplicon. The reactions were performed in a 50 μL reaction mixture volume, containing 10 ng of template gDNA (2–4 μL), 0.2 μM of each primer (1 μL of 10 μM), and 25 μL of 2× PCR Master Mix (KOD One™ PCR Mastermix; Toyobo, Japan). The PCR conditions were set as follows: initial denaturation at 94 °C for 2 min, 45 cycles of amplification (94 °C for 30 s, 55 °C for 30 s, and 68 °C for 45 s), and a final extension at 72 °C for 5 min. The reaction was performed using a T100™ Thermal Cycler (Bio-Rad, Hercules, CA, USA). The DW and gDNA of *A. caninum* and *A. ceylanicum* were used as negative and positive controls, respectively. The PCR products were purified using a NucleoSpin^®^ PCR Clean-up Kit (Macherey-Nagel GmbH, Düren, Germany). Purified PCR products were submitted for fluorescent dye-terminator sequencing by Bio Basic, Inc. (Singapore, Singapore) in the sense and antisense directions using the PCR primer set described above. For nucleotide analysis, the *cox1* amplicons of 32 samples (29 *A. ceylanicum* samples from dogs, 2 *A. ceylanicum* samples from soil, and 1 *A. caninum* sample from dog) were selected based on the geographic distribution, covering all positive provinces.

### 2.5. Data Analyses

#### 2.5.1. Prevalence of Hookworm Infection/Contamination

Positive fecal samples from the zinc sulfate flotation technique and positive soil samples from the Baermann technique were considered as positive for hookworm detection. The prevalence of hookworm infection in dogs and contamination in soil samples were estimated using the formula prevalence = (number of positives samples divided by the number of samples in dogs/soil examined) × 100. The association between hookworm-positive results and sampling areas was assessed using Fisher’s exact test. The significance was defined as *p* < 0.05. 

#### 2.5.2. Phylogenetic Analysis

The *cox1* sequences were edited and manually aligned. The consensus sequences were generated using the BioEdit Sequence Alignment Editor [34] and then compared with the nucleotide sequences from GenBank using a Basic Local Alignment Search Tool (BLAST) analysis. Phylogenetic analysis was performed using MEGA X [35]. Multiple sequences were aligned using ClustalW (https://www.genome.jp/tools-bin/clustalw), and phylogenetic analysis was performed using a maximum likelihood (ML) method based on the Kimura 2-Parameter model. The consensus tree was obtained after a 1000-replication bootstrap analysis. The *cox1* sequences of *A. ceylanicum* from other areas of Thailand and other countries, including Malaysia, Cambodia, China, Japan, Papua New Guinea, Australia, and Tanzania, and *A. caninum* from Australia (NC012309) were used as reference sequences for tree construction (Appendix A). In addition, the sequence of *A. duodenale* from China (accession number: NC003415) was used as an outgroup species. 

#### 2.5.3. Population Genetic Analysis

The DnaSP 6 program [36] was used for the haplotype analysis and calculations to determine the haplotype diversity (Hd), nucleotide diversity (π), and the number of variable sites (S). The gene flow among populations was analyzed by fixation index (F_ST_) statistics using the Arlequin program Version 3.5 [37]. The gene flow of *A. ceylanicum* was analyzed using the reference sequences from Asian countries, including Thailand, China, Malaysia, and Cambodia (Appendix A). The Network program [38] was used to generate a median-joining (MJ) network of *cox1* haplotypes and reference haplotypes and was subsequently drawn manually. The references haplotype and their frequency obtained from the NCBI database (https://www.ncbi.nlm.nih.gov/genbank/) are described above. 

## 3. Results

### 3.1. Prevalence and Distribution of Hookworm Infection

A total of 299 fecal samples and 212 soil samples from 53 temples were examined microscopically, and the hookworm species were identified by PCR-RFLP targeting the ITS region. The prevalence rate of hookworm in dogs was 26.4% (79/299), as assessed by microscopic examination. PCR-RFLP successfully identified the species of hookworm in 58 samples, and the rates of *A. ceylanicum* and *A. caninum* infection were 96.55% (56/58) and 3.45% (2/58), respectively (Figure 1). On the other hand, the prevalence of hookworm-like larvae in soil was 10.38% (22/212), as assessed by microscopic examination (Table 1). However, PCR-RFLP only identified the species in 8 of 22 samples, and all the samples were identified as *A. ceylanicum* (Table 1). 

Considering the population, the temple-level prevalence of positive dogs and positive soils was 47.17% (25/53) and 26.42% (14/53), respectively (Appendix A). A total of 10 temples were positive for hookworm both in dog and soil samples (Appendix A). Although not statistically significant, the hookworm-positive results in dogs were speculated to be associated with the hookworm-positive results in the soil (*p* = 0.06). Regarding the different areas in each temple, we found that the soil from dog-dwelling areas had the highest rates of hookworm contamination (15.09%, 8/53), followed by the soil from human activity areas, such as sand pagoda (11.32%; 6/53); the areas under big trees (9.43%; 5/53); and the temple courtyard (5.66%, 3/53) (Appendix A). However, the frequency of hookworm-contaminated soil in each sampling site in the temple exhibited no significant differences (*p* > 0.05; Fisher’s exact). 

### 3.2. Hookworm Species Confirmation and Genetic Characteristics of the A. ceylanicum Mitochondrial cox1 Gene

The nucleotide BLAST of partial *cox1* sequences (315 bp) of 31 *A. ceylanicum* sequences exhibited a high identity (98.07–100%) with Thai *A. ceylanicum* (GenBank accession no. KF896595). An *A. caninum* sequence from a dog exhibited a 96.71% identity with *A. caninum* from Australia (GenBank accession no. AJ407962). The obtained *A. ceylanicum cox1* gene sequences from dog and soil isolates were classified into nine haplotypes, which were referred to as Acy-COX1-TH01 to Acy-COX1-TH09. The sequences of *A. ceylanicum* of dogs and soil were deposited in GenBank (DDBJ/EMBL/GenBank database accession no. LC533318-LC533327). The sequence of *A. caninum* was deposited as LC533328 (Appendix A). 

Genetic variation among nine haplotypes of Thai *A. ceylanicum*, as assessed by multiple alignment analysis, revealed 12 different variable sites, including 2 transversions and 10 transitions (Table 2).

The distribution and frequencies of *A. ceylanicum* haplotypes in Upper Northern Thailand were analyzed and revealed that the Acy-COX1-TH01 haplotype was detected at the highest frequency at 74.2% (23/31 sequences). This haplotype was commonly observed in all four provinces and found in both fecal and soil samples. As displayed in Figure 2, the sequence of two soil isolates (Acy-COX1-TH01-soil) was close to Acy-COX1-TH01-dog. Six haplotypes are present in Chiang Mai (Acy-COX1-TH01 to TH06), 3 haplotypes are present in Lampang (Acy-COX1-TH01, TH08, and TH09), whereas only one or two haplotypes were observed in Chiang Rai (Acy-COX1-TH01 and TH07) and Phayao (Acy-COX1-TH01) provinces (Appendix A). Regarding the relationship among the human and animal isolates, Thai Acy-COX1-TH01 was close to the sequences of *A. ceylanicum* from Malaysian people (MK792828), dog (MK792824), and cat (MK792825) that belong to the same haplotype MA02 (Figure 2 and Appendix A). Similarly, Acy-COX1-TH03 was close to the sequences of *A. ceylanicum* from human (KC247737) and dog (KC247734) that belong to the haplotype Malaysian MA01. Regarding the relationship between *A. ceylanicum* and geographic distribution, a phylogenetic tree demonstrated that *A. ceylanicum* from Thailand and all reference countries, including Malaysia, Cambodia, China, Japan, Papua New Guinea, and one country from the African continent (Tanzania), were located in the same clade, with the exception of one human sequence in Australia (AJ407937) (Figure 2). 

### 3.3. Population Analyses of the A. ceylanicum Mitochondrial cox1 Gene among Asian Countries 

The haplotype network displayed a star-like pattern and exhibited a wide dispersal around the predominant haplotype Acy-COX1-TH01 (in this study). This haplotype was identical to haplotypes MA02 from Malaysia, CA01 from Cambodia, and CH01 haplotype from China (Figure 3). The second most abundant haplotype was Acy-COX1-TH03, which was placed in the same group as haplotype MA01 from Malaysia and haplotype CA03 from Cambodia. Acy-COX1-TH06 was the most distinct haplotype, which includes four nucleotide-substitutions and was delineated from the haplotype CA02 from Cambodia. 

The genetic diversity of the Thai *A. ceylanicum* population has been analyzed and compared pairwise to the populations of Cambodia, Malaysia, and China. The Thai *A. ceylanicum cox1* population (32 sequences; 31 from this study and 1 sequence from Inpankaew et al. [33]) displayed the lowest genetic diversity—i.e., a haplotype diversity (Hd) of 0.4435 and a nucleotide diversity (π) of 0.0036 (Table 3). 

The highest haplotype diversity and nucleotide diversity were observed in the Chinese *A. ceylanicum* population (Hd = 0.9394; π = 0.0201), followed by the Cambodian and Malaysian populations. Pairwise (F_ST_) indices between the Thai *A. ceylanicum* population and other reference populations were 0.2644, 0.2507, and 0.1526 for the Chinese, Cambodian, and Malaysian populations, respectively. In addition, the highest F_ST_ value of 0.3577 was observed between the Chinese and Malaysian populations. All the F_ST_ values were statistically significant (Table 4).

## 4. Discussion

The major hookworm species in the community dogs in Upper Northern Thailand was *A. ceylanicum*. We demonstrated an approximately 50% temple-level prevalence of hookworm infection in dogs, and these positive dogs were commonly living in the community temples and freely roaming in the surrounding communities. The prevalence of hookworm infection at the individual dog level in this study (26.4%) showed an approximately two-fold increase compared with previous studies in Chiang Mai province (approximately 12%), as reported by Tiwananthagorn et al. (unpublished results) and Tangtrongsup et al. [23], and greater than that reported by Pumidonming et al. [22] in Lower Northern Thailand (21.3%). These differences are probably due to the different study areas and the inclusion criteria of no regular deworming program. The proportion of *A. ceylanicum* infection in dogs from our study in Upper Northern Thailand (96.6%) was higher than the previous report in the Lower Northern region, which reported a rate of 82.1% [22]. Recently, various studies have reported the rate of *A. ceylanicum* in hookworm infection in the bordering countries of Thailand: 94.4% of dogs in Cambodia, 52% of dogs in Malaysia [39], 77.8% of dogs in Laos [40], and 42.7% of dogs in China [7]. The failure of PCR amplification in some fecal and soil samples was noticed in the present study and was also noted in other studies. This failure could be due to the low intensity of infection, resulting in a reduced number of hookworm eggs, and some PCR inhibitors in the fecal and soil samples interfere with the amplification reaction [15,41]. Therefore, developing a more robust PCR assay to improve the diagnosis of hookworm infection is needed for the efficient control of this zoonosis.

The temples with hookworm-infected dogs are prone to have hookworm-contaminated soil. Here, we confirmed the presence of *A. ceylanicum* larvae in the soil. Moreover, the *A. ceylanicum* genotypes in dogs and soil were identical, suggesting the shedding of hookworm from dogs to the soil. The frequently detected areas in the positive temples included the dog’s dwelling areas; human activity areas, such as the sand pagoda; and the areas under the big tree. These areas are the places that community people commonly access for the Buddhist rituals, ceremonies, commerce, and recreation. In addition, allowing dogs to roam freely with indiscriminate defecating areas in the Thai temples can lead to the widespread contamination of infective *A. ceylanicum* hookworm larvae in the environment and possible transmission to humans who occasionally go barefoot during these activities. As suggested in rural Malaysia, close contact with community dogs and cats was associated with *A. ceylanicum* hookworm infection in humans [42]. Previous reports worldwide have reported soil contaminated with hookworm in the public parks of Brazil and Malaysia [43,44], a tribal area of India [5], dog parks in Portugal [45], and organic farms in the Philippines [46]. Therefore, good hygienic practices of shoe-wearing and temple ground cleaning programs should be encouraged to reduce the risk of transmission [47]. In addition, reducing the number of free-roaming dog population size through a sterilization campaign or by sheltering free-roaming dogs should be implemented [48].

The molecular epidemiologic data, phylogenetic relationship, and population analysis obtained from the characterization of the *cox1* gene of *A. ceylanicum* hookworms strongly support the distribution of this parasite in the country, within the Asian continent, and possibly worldwide. A common haplotype of Thailand (Acy-COX1-TH01), which was observed in all four provinces of Upper Northern Thailand, suggested the circulation of *A. ceylanicum* in the region. Genetic differentiation analysis showed that Thai *A. ceylanicum* had the lowest haplotype diversity and nucleotide diversity, indicating minimal differences between haplotypes in Thailand. However, further studies in other regions of Thailand are necessary. The pairwise fixation index showed that Thai *A. ceylanicum* was mostly close to Malaysian *A. ceylanicum*, suggesting gene flow between these populations. In addition, the sharing haplotype of the *cox1* gene of *A. ceylanicum* among the countries (Acy-COX1-TH01 and MA02 and CA01 and CH01; or Acy-COX1-TH03 and MA01 and CA03) advocated the transmission and widespread geographic distribution among these countries [49]. The haplotype relationship between Malaysia and Cambodia populations revealed a very complex evolutionary pattern with some median vectors that indicated many lost or potential ancestors [50]. Interestingly, the Chinese *A. ceylanicum* exhibited the highest haplotype and nucleotide diversities, suggesting that the Chinese *A. ceylanicum* might be an ancestor of the populations in Asian countries. Our hypothesis is consistent with a previous study on the history of domestic dogs, in which China was the origin of the dog species [51]. Animals potentially migrated to accompany human colonization to other Asian countries and subsequently carried the parasites and allowed the parasites to colonize new habitats [51,52,53]. Hence, the transportation of companion animals should strictly follow the guidelines and regulations of each county to prevent the spreading of zoonotic hookworm.

Based on the phylogenetic tree, the sequences of *A. ceylanicum* from humans (MK792828-29), dogs (MK792824), and cats (MK792825) in Malaysia belong to the same haplotype MA02, confirming that transmission from dogs to humans has occurred. This species was also found in wildlife, such as Asian golden cat, leopard cat, civet, and wild canids [9,54,55]. The biological, epidemiological, and pathophysiological characteristics of *A. ceylanicum* in humans and other wildlife species in Thailand and their genetic diversity with isolates from companion animals deserve further investigation. 

Integrated control programs intended for combining chemotherapeutic interventions by deworming programs with improvements in community hygiene and animal health programs will aid in curbing this potentially opportunistic zoonosis. For the deworming program of community dogs in Thailand, an oral dose of 10 mg/kg BW of pyrantel pamoate and the subcutaneous injection of 0.2–0.4 mg/kg BW ivermectin have been regularly administered for deworming; however, the efficacy of these regimens against *A. ceylanicum* in dogs remains unclear. Currently, many commercial drugs against *A. ceylanicum* infection in dogs have been tested at the laboratory level and exhibited greater than 99% efficacy within 3–7 days, including a combination of 15 mg/kg BW of febantel and 14.4 mg/kg BW of pyrantel embonate (Drontal^®^ Plus, Bayer Animal health) [56], a spot-on of 2.5 mg/kg BW moxidectin (Advocate^®^, Bayer Animal health) [57], and 0.5 mg/kg BW milbemycin oxime (NexGard Spectra^®^, Boehringer Ingelheim) [58]. The clinical field assessment of the efficacy of anthelminthic agents against *A. ceylanicum* infection warrants further investigation.

## 5. Conclusions

In conclusion, this study provided the data, genetic characteristics, and evolutionary relationship of *A. ceylanicum* in Thailand and other Asian countries. We confirmed that *A. ceylanicum* was the predominant hookworm species in Northern Thailand, and the data fully supported the transmission dynamics from dogs to soil environments. Therefore, the awareness of *A. ceylanicum* transmission from dogs and soil in the temple should be raised. A proper schedule for the deworming of community dogs and the regular cleaning of the temple ground are suggested to reduce the risk of transmission of this zoonotic disease.

## Figures and Tables

**Figure 1 animals-10-02154-f001:**
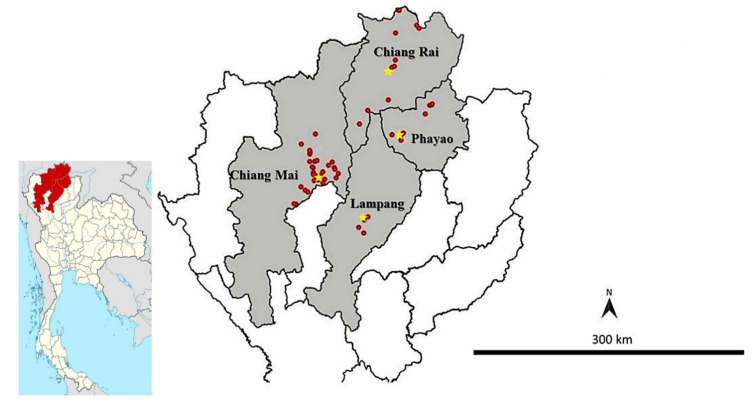
Map of the study areas in Thailand. The gray color indicates four study provinces, red dots indicate the location of sampling sites (temples), and stars indicate the capital city of each province.

**Figure 2 animals-10-02154-f002:**
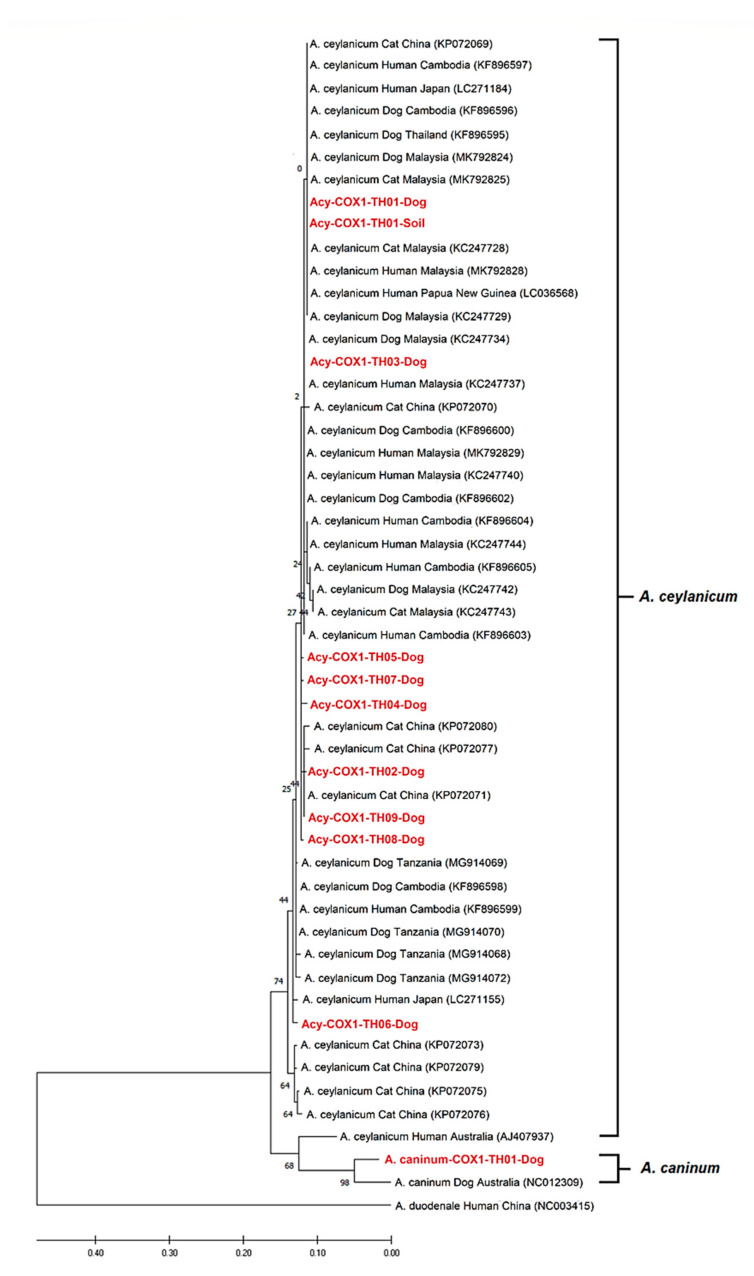
The evolutionary relationship of *A. ceylanicum* and *A. caninum* isolates from dogs and soil based on the 259-bp fragments of the mitochondrial cytochrome *c* oxidase subunit 1 (*cox1*) gene. The tree was constructed using a maximum likelihood method based on the Kimura 2-Parameter using the MEGA X software version 10.0.5. The number in each branch indicates the percentage of 1000 bootstrap replications. Sequences obtained from GenBank are indicated by their accession numbers.

**Figure 3 animals-10-02154-f003:**
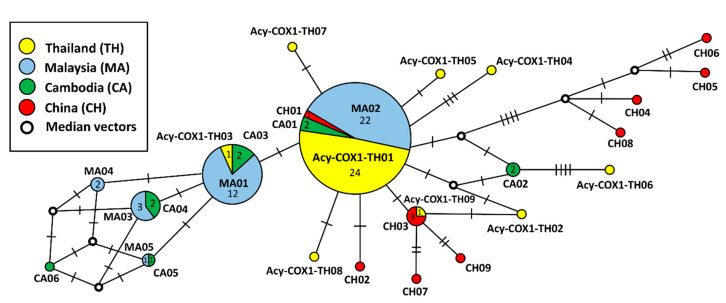
The haplotype network map of *A. ceylanicum* based on a fragment of the *cox1* gene from Thailand, Cambodia, China, and Malaysia. Circle size is scaled to frequencies of each haplotype. The hash marks indicate nucleotide substitutions among adjacent haplotypes.

**Table 1 animals-10-02154-t001:** Summary of hookworm infection in dog fecal samples and soil samples in Thailand, as demonstrated by the microscopic results and subsequent PCR-RFLP targeting the ITS region.

Province	Fecal Samples									Soil Samples								
	Positive Microscopic			N_2_	*A. ceylanicum*		*A. caninum*		n/a	Positive Microscopic			N_4_	*A. ceylanicum*		*A. caninum*		n/a
	N_1_	*n*	%		*n*	%	*n*	%	*n*	N_3_	*n*	%		*n*	%	*n*	%	*n*
Chiang Mai	159	57	35.85 ^a^	42	40	95.24	2	4.76	15	120	16	13.33	3	3	100	0	0	13
Chiang Rai	70	6	8.57 ^b^	5	5	100	0	0	1	48	1	2.08	1	1	100	0	0	0
Lampang	20	9	45.00 ^a^	5	5	100	0	0	4	16	3	18.75	2	2	100	0	0	1
Phayao	50	7	14.00 ^b^	6	6	100	0	0	1	28	2	7.14	2	2	100	0	0	0
Total	299	79	26.42	58	56	96.55	2	3.45	21	212	22	10.38	8	8	100	0	0	14

N_1_: total number of examined dog fecal samples assessed by microscopic examination; N_2_: number of positive dog samples successfully assessed with PCR-RFLP; N_3_: total number of examined soil samples by microscopic examination; N_4_: number of positive soil samples successfully assessed with PCR-RFLP; n/a: species unidentified samples; superscripts (a,b) indicate test results of Chi-square on prevalence among four provinces (any measurements with shared superscript letters are not significantly different from each other at *p* < 0.05).

**Table 2 animals-10-02154-t002:** Multiple alignment of the partial *cox1* from 9 haplotypes of *A. ceylanicum* in this study.

Haplotype Name	Nucleotide Position on the *A. ceylanicum cox1* Gene											
	48	72	87	88	90	105	147	150	198	199	219	264
Acy-COX1-TH01	A	C	A	G	A	A	G	A	G	G	A	G
Acy-COX1-TH02	G	.	.	.	.	.	A	.	.	.	.	.
Acy-COX1-TH03	.	T	.	.	.	.	.	.	.	.	.	.
Acy-COX1-TH04	.	.	.	.	.	G	.	.	A	A	.	.
Acy-COX1-TH05	.	.	.	C	.	.	.	.	.	.	.	.
Acy-COX1-TH06	.	T	G	.	G	.	A	.	.	.	G	A
Acy-COX1-TH07	.	.	.	.	.	.	.	T	.	.	.	.
Acy-COX1-TH08	.	.	G	.	.	.	.	.	.	.	.	.
Acy-COX1-TH09	G	.	.	.	.	.	.	.	.	.	.	.

Dots indicated the matching of nucleotide position the *A. ceylanicum* sequence.

**Table 3 animals-10-02154-t003:** Summary of the genetic diversity of the 4 populations of *A. ceylanicum* based on the nucleotide sequences of the partial mitochondrial *cox1* gene.

Populations	N	Diversity			
		S	h	Hd ± SD	π ± SD
Thailand	32	11	9	0.4435 ± 0.1105	0.0036 ± 0.0028
Cambodia	10	6	6	0.9111 ± 0.0620	0.0088 ± 0.0059
Malaysia	40	4	5	0.6141 ± 0.0593	0.0043 ± 0.0006
China	12	17	9	0.9394 ± 0.0577	0.0201 ± 0.0118

N: numbers of *A. ceylanicum* sequences used; S: numbers of variable sites; h: numbers of haplotypes; Hd: haplotype diversity; SD: standard deviation; π: nucleotide diversity.

**Table 4 animals-10-02154-t004:** The pairwise fixation index (F_ST_) among four *A. ceylanicum* populations based on the partial mitochondrial *cox1* gene.

Populations	Thailand	Cambodia	Malaysia	China
Thailand	-			
Cambodia	0.2507	-		
Malaysia	0.1526	0.0999	-	
China	0.2644	0.1939	0.3577	-

All values were statistically significant (*p* < 0.001).

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
