# Peer review of "Ancylostoma ceylanicum*: The Neglected Zoonotic Parasite of Community Dogs in Thailand and Its Genetic Diversity among Asian Countries"

_animals, 2020, doi:10.3390/ani10112154_

Round 1
Reviewer 1 Report
The manuscript described the detailed information of Ancylostoma ceylanicum prevalence in dogs of northern Thailand that has not been clarified in detail yet. Therefore, I judged it would be of worth for publishing in this Journal, however, there are some points to be improved before publishing. The followings are my comments made on this manuscript.
L30, L38 and others: There are many sentences with beginning "A. ceylanicum ....". Please make sure that it is acceptable in this Journal to begin a sentence with abbreviated word such as "A.".
L30-31 and L38: The sentence "A. ceylanicum was the main hookworm ..." was repeated, thus redundant.
L33 L82 and others: "cox1" should be italic (gene name should be italic). Check this for whole text, Tables and Figures.
L56: A. of A. caninum should be full-spelled.
L57. U. of U. stenocephala should be full-spelled.
L59: Re-consider to use "However". "Among Ancylostoma species," would be fit for this.
L62-63: From what evidence the sentence was made?
L62, L93 and others: "Ancylostoma" should be italic.
L66: Re-consider to use "Interestingly".
L79-80: Ref 17 is not for Ancylostoma, thus this sentence is not appropriate.
L81: "c" of cytocrome c should be italic. "sbuunit1" should be "subunit 1".
L85 to94: The information of the prevalence and prevalent species in humans in the study region should be included in this paragraph.
L98-99: Although several study sites were allocated in each province, only one coordinate point was described for each province in the text. Where does each coordinate point show?
L102: (i) having a high human population ____ What is the criteria for "high"?
Figure 1. The map of whole Thailand should be larger.
L118: "fresh fecal samples" should be "fresh feces".
L160 and others: There are several "A. caninum" in non-italic in the text. Those should be written in intalic.
L166, L167, L170: A. ceylanicum, A. caninum, cox1 should be italic.
L218-221 and Table 1: There are number calculations expressed with one or two decimal places. Those should be described by the same regulation considering significant firures (digits).
Table 1: The table require wider column width.
L232: Re-consider for using "population" in "the population of 53 temples".
L235: How could you say results in dogs were associated with the results in the soil under statistically non-significant base? It may be speculated but should not be defined.
L242: A. ceylanicum, cox1 should not be italic.
Table 2: The haplotypes names used in Table 2 are not matched to the haplotypes names used in Table S2. Those should be matched.
Figure 2: Why did you explain Acy in the foot-note? Acy is the part of the name of hplotypes, isn't it? Put space between A. and ceylanicum, caninum, duodenale for registered sequence.
L279: "haprotypes network" should be "haprotype network"
L280: Re-consider for use of "similar".
Figure 3: It is better to use color for the haplotypes reported from different countries.
Also names of the haplotypes that you reported in this study are different from those in Table 2.
L309-310: Reconsider the sentence with carful check of its grammar.
Author Response
Your comments and suggestions are greatly appreciated. The followings are the point-by-point response to the comments.
"Please see the attachment"
Saruda Tiwananthagorn, DVM, Ph.D. Assistant Professor
Corresponding author
Department of Veterinary Biosciences and Veterinary Public Health,
Faculty of Veterinary Medicine, Chiang Mai University,
Mae Hiae, Muang, Chiang Mai 50100, Thailand
Telephone:+6653-948-046
Mobile: +6695-446-5955; Fax: +6653-948-065
E-mail: saruda.t@cmu.ac.th

Reviewer 2 Report
In this paper, the authors attempted to confirm the hookworm species in dogs and soil samples collected from different areas of Thai temples and also investigate the genetic diversitiy of Ancilostoma ceylanicum among the Thai hookworm population and the population of neighboring Asian countries.
The main contribution of this study is the identification, for the first time, of hookworm species that affect dogs in the Northern Thailand.
The paper highlights the importance of knowing the prevalence of hookworm in endemic areas and looking for solutions to reduce the risk of transmission to humans.
The paper is well written and organized. Each topic is evaluated in different paragraphs and is well raised and discussed.
Specific comments:
Introduction
Line 56: Please, specify the full form of “Ancylostoma” at the first time you cite it in each section of the manuscript.
Line 57: As above, please specify the full form of “Uncinaria”.
Line 66: Specify which hosts.
Materials and methods
Line 131: Specifty the sample cooling temperature. 4ºC?
Line 144: What does “Buffer SE” mean? I suppose that it is the elution buffer provided with the kit, but clarify.
Line 156: You specify the concentration of the reagents that make up the PCR reaction, but also specify the volume to which each reagent is added to the PCR reaction (gDNA, primers and Master Mix).
Line 160: Write “A. caninum” in italics.
Line 161: Use the same symbol to indicate degrees throughout the text.
Line 172: Indique the reagent volumes.
Line 191: Write “p” or “P” evenly throughout the text.
Line 199: Write “A. ceylanicum” in italics.
Results
Lines 244-245 and 249: Write “A. caninum” in italics.
Line 291: Write “A. ceylanicum” in italics.
Discussion
Line 324: I am struck by the lower percentage of positives detected by PCR compared to microscopy. The authors point out a failure in the PCR due to low intensity of infection or by the presence of PCR inhibitors. Maybe the authors could indicate here the need to develope or pursue more robust PCR assays to improve the control of this zoonosis.
Line 340: the authors indicate that good hygiene practices such as shoe-wearing and cleaning programs in temples would help to decrease the risk of transmission. I agree. But I think an effective measure would be to reduce the number of dogs that roam freely in human settings, especially areas with high public influx. I suggest including a brief discussion on that.
Author Response

(The authors gave the same response as above.)
